



# Homogenization of the Observatoire de Haute Provence ECC ozonesonde data record: comparison with lidar and satellite observations

Gérard Ancellet[1], Sophie Godin-Beekmann[1], Herman G.J. Smit[2], Ryan M. Stauffer[3], Roeland Van Malderen[4], Renaud Bodichon[5], and Andrea Pazmiño[1]

[1]LATMOS, Sorbonne Université, Université Versailles St-Quentin, CNRS/INSU, Paris, France
[2]Forschungszentrum Jülich IEK-8, Jülich, Germany
[3]NASA Goddard Space Flight Center, Atmospheric Chemistry and Dynamics Lab, Greenbelt, Md, USA
[4]Royal Meteorological Institute, Uccle, Belgium
[5]IPSL, Sorbonne Université, Université Versailles St-Quentin, CNRS/INSU, Paris, France

**Correspondence:** gerard.ancellet@latmos.ipsl.fr

**Abstract.** The Observatoire de Haute Provence (OHP) weekly Electrochemical Concentration Cell (ECC) ozonesonde data have been homogenized for the time period 1991-2020 according to the recommendations of the Ozonesonde Data Quality Assessment (O3S-DQA) panel. The assessment of the ECC homogenization benefit has been carried out using comparisons with ground based instruments also measuring ozone at the same station (lidar, surface measurements) and with collocated satellite observations of the $O_3$ vertical profile by Microwave Limb Sounder (MLS). The major differences between uncorrected and homogenized ECC are related to a change of ozonesonde type in 1997, removal of the pressure dependency of the ECC background current and correction of internal ozonesonde temperature. The 3-4 ppbv positive bias between ECC and lidar in the troposphere is corrected with the homogenization. The ECC 30-years trends of the seasonally adjusted ozone concentrations are also significantly improved both in the troposphere and the stratosphere when the ECC concentrations are homogenized, as shown by the ECC/lidar or ECC/surface ozone trend comparisons. A -0.29% per year negative trend of the normalization factor ($N_T$) calculated using independent measurements of the total ozone column (TOC) at OHP disappears after homogenization of the ECC. There is however a remaining -5% negative bias in the TOC which is likely related to an underestimate of the ECC concentrations in the stratosphere above 50 hPa as shown by direct comparison with the OHP lidar and MLS. The reason for this bias is still unclear, but a possible explanation might be related to freezing or evaporation of the sonde solution in the stratosphere. Both the comparisons with lidar and satellite observations suggest that homogenization increases the negative bias of the ECC up to 10% above 28 km.

## 1 Introduction

Stratospheric ozone recovery is expected due to a decrease of ozone depleting substances. Trends of ozone in the upper troposphere, lower and mid stratosphere however show latitudinal and seasonal variabilities which depend on (i) dynamical variability of the atmosphere, (ii) the temperature dependence of stratospheric ozone photochemistry, (iii) the increase of





tropospheric ozone precursors in the upper troposphere (Szelag et al., 2020; Cohen et al., 2018; Thompson et al., 2021). A large number of validation and intercomparison studies of free tropospheric and lower stratospheric ozone use balloon borne Electrochemical Concentration Cells (ECC) as reference (Tarasick et al., 2021). At Observatoire de Haute Provence (OHP), stratospheric and free tropospheric ozone monitoring is carried out since the mid-1980s with ozonesonde and lidar observations. The OHP station located at 44°N,6°E is one of the few long-term measuring stations for vertical ozone profiles in Southern Europe. This station allows to characterize (i) the impact of ozone sources observed in one of the hot spots of the tropospheric ozone columns observed by satellite (Richards et al., 2013) and (ii) the climate variability on mid-latitudes total column ozone (Zhang et al., 2015; Petkov et al., 2014). Improvement and homogenization of the OHP ozone ECC observations have been achieved from 1991 to 2021 using the recent ozonesonde data quality assessment (O3S-DQA) panel recommendations (Smit et al., 2012). An extensive use of lidar measurements both at tropospheric and stratospheric altitudes together with co-located satellite observations obtained during the OHP ECC soundings has allowed the quantification of the ozone measurement accuracy improvement achieved with this homogenization of the ECC ozonesondes. Sections 2 and 3 summarize the corrections made to the ozonesonde measurements and the methodology for assessing its benefit. Section 4 presents and discusses the results of the different instrumental comparisons and the changes obtained in terms of interannual ozone variation at different altitudes between 0.7 and 30 km ASL.

## 2 Description of the ozonesonde homogenization

A total number of 1412 ECC ozonesondes have been launched at OHP since 1991 when we replaced Brewer-Mast regular soundings by ECC sondes following the preparation instructions of Komhyr (1986) just after a lidar/ozonesonde intercomparison campaign held at OHP in 1989 (Beekmann et al., 1994). Ozonesondes are launched once a week generally near 9 UT but 40 soundings have been made during the night either for lidar/ozonesonde comparison or for detection of long range transport of polar ozone streamers forecasted by chemical transport model models. The ozone partial pressure $P_{O3}$ measured in mPa by the ECC is related to the electrochemical current I measured in $\mu A$, the background current $I_b$ measured in the preparation laboratory with an ozone removal filter after the sonde being exposed to ozone, the internal temperature of the air sample $T_i$ in K, the capture efficiency of the $O_3$ in the liquid phase $\alpha$, the stoichiometry S of the $O_3$ to $I_2$ conversion and the ECC pump flow rate, $\phi_p$ in $cm^3 s^{-1}$ (Smit and Thompson, 2021).

$$P_{O3} = 4.307\,10^{-2}.(I - I_b).T_i\,/\,(S.\alpha.\phi_p) \tag{1}$$

A major change in the sounding procedure occurred in 1997 when the Science Pump Corporation (SPC) ozonesonde was replaced by an EnSci ozonesonde while using a Sensing Solution Type (SST) of 1% which means 1% KI concentration and a full buffer concentration (Smit and Thompson, 2021). Using the instructions given by the O3S-DQA, the following corrections have been implemented before a new calculation of $P_{O3}$ after homogenization using equation 1:

- Change of $\alpha$ and its pressure dependency before 1996 when 2.5 $cm^3$ of KI solution was used in the cell.





- Scaling of $P_{O3}$ measured by the EnSci-SST 1% ozonesondes after March 1997 to $P_{O3}$ from SPC-SST 1% ozonesonde observations made before March 1997 assuming that SPC-SST 1% is a better reference than EnSci-SST 1% (Deshler et al., 2017). This correction is larger than -10% at altitudes above 30 km and of the order of -4% in the troposphere.

- When $I_b > 0.1$ $\mu$A (less than 6% of the data set), $I_b$ is replaced by 0.05 $\mu$A, the average of the measured background current for our dataset, while the uncertainty of $I_b$ becomes 0.1 $\mu$A.

- The pressure dependency of the background current has been removed for the homogenized version since the $O_2$ concentration is not playing a significant role in the residual current when ozone is removed (Thornton and Niazy, 1983; Vömel and Diaz, 2010).

- No more vertical smoothing of the ozone partial pressure

- Correction of measured $T_i$ to account for changes in the position of the thermistor and for differences with the true air sample temperature (the thermistor was tapped to the pump before July 2007 and inserted in the pump hole since that time)

- Correction of the pump flow rate to account for humidification effect when using the bubble flowmeter to determine the
65 flowrate in the laboratory as part of the pre-flight preparation of the sonde.

- Two different pump flow rate efficiency correction tables at pressure below 100 hPa are now applied for EnSci (Komhyr95) and SPC (Komhyr86) ozonesonde. Only Komhyr86 is applied for the current to $P_{O3}$ conversion of uncorrected data.

As the background current uncertainty is a significant contribution to the $P_{O3}$ uncertainty in the upper troposphere (Van Malderen et al., 2016), the comparison of $I_b$ used before and after homogenization is shown in Fig.1. We see that there is no significant
trend of the background current between 1991 and 2021 and only 17% of $I_b$ values are above 0.05 after homogenization. The removal of the pressure dependency of the background current leads to significant relative differences (>5%) in the upper troposphere where the ECC current is smallest. In February 2004, the UHF receiver and ground calibration tools have been changed and a new processing software STRATO developed by H. Vömel at NOAA (http://cires1.colorado.edu/ voemel/strato/strato.html) was implemented for the $O_3$ partial pressure retrieval. The raw ECC current data files no longer exist between February 2004
and July 2007 and the ECC current have been retrieved from the uncorrected ozone partial pressure using equation 1. The recorded lab temperature and relative humidity are taken for the pump flow rate correction of ozonesondes launched after June 1999 (except in 2002 and 2003), for the other dates the monthly means of the lab temperature and relative humidity terms in the pump flow rate equation (Smit et al., 2012) are used.

In July 2007 the radiosonde was changed switching from Vaisala RS80 to MODEM M10. The MODEM M10 measures
the true GPS altitude with a pressure altitude retrieved from this measurement. No correction is applied on the RS80 pressure measurements and an offset of 0.5-1 hPa may exist in the stratosphere pressure above 20 km before 2007 (Tarasick et al., 2021; Stauffer et al., 2014).

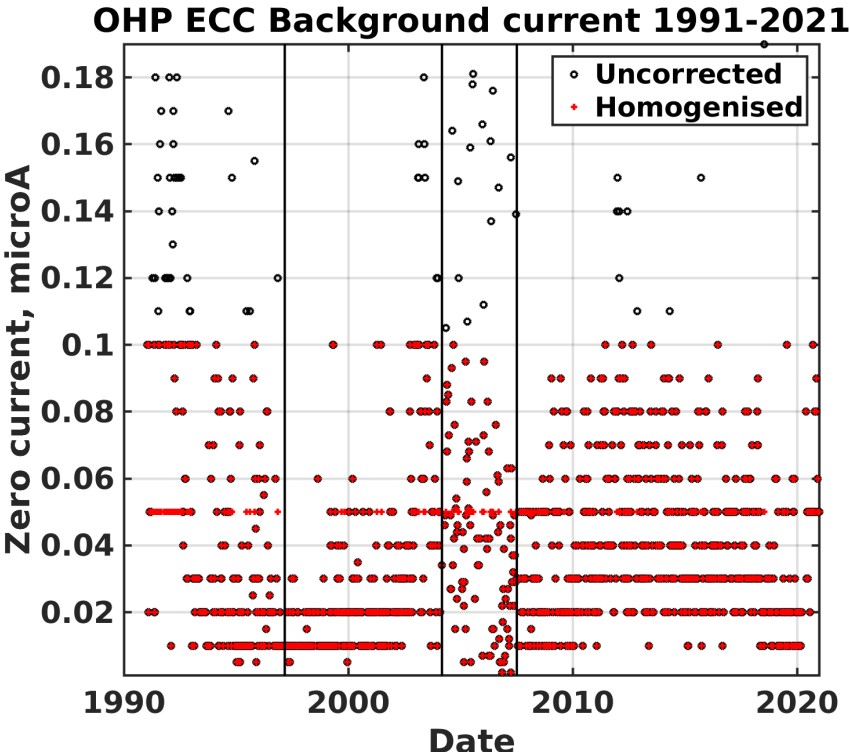

**Figure 1.** Time evolution of the OHP ECC background current from 1991 to 2021. The red crosses are the currents after homogenization and the black circle before. The black lines correspond to the major ozonesonde changes in 1997, 2004 and 2007 (see Fig. 2).

The homogenized minus uncorrected ECC partial pressure differences normalized to the homogenized ECC partial pressure are shown in Fig.2. Significant overall negative differences ($\leq$ -5%) are obtained (i) in the upper troposphere (8-12 km) because of removal of $I_b$ pressure dependency and (ii) above 28 km after 1997 when taking into account the change to EnSci. Positive differences reaching 5% in the stratosphere are also observed for the SPC period before 1997 because of the positive pump flow rate (+2%) and ECC temperature (+3%) corrections without any negative corrections in the stratosphere.

SAOZ (Systéme d'Analyse par Observation Zénithale) and Dobson UV/visible spectrophotometer total ozone column (TOC) measurements are also available at OHP. So a so called normalization factor ($N_T$) is calculated as the ratio of the spectrophotometer TOC and the ECC TOC. The TOC corresponding to the ECC soundings is calculated using the integration of the ozone concentrations up to the burst altitude when it is higher than 25 km. The residual ozone above the burst altitude has been calculated using a monthly mean climatology based on the OHP stratospheric lidar up to 40 km and satellite observations above (Robbins et al., 1989). The residual ozone is also multiplied by the ratio between the climatological $O_3$ concentration at burst level and the measured ECC $O_3$ concentrations in the 100-m layer below the burst level so that only the relative dependence of the monthly climatology with altitude is taken into account in this calculation.

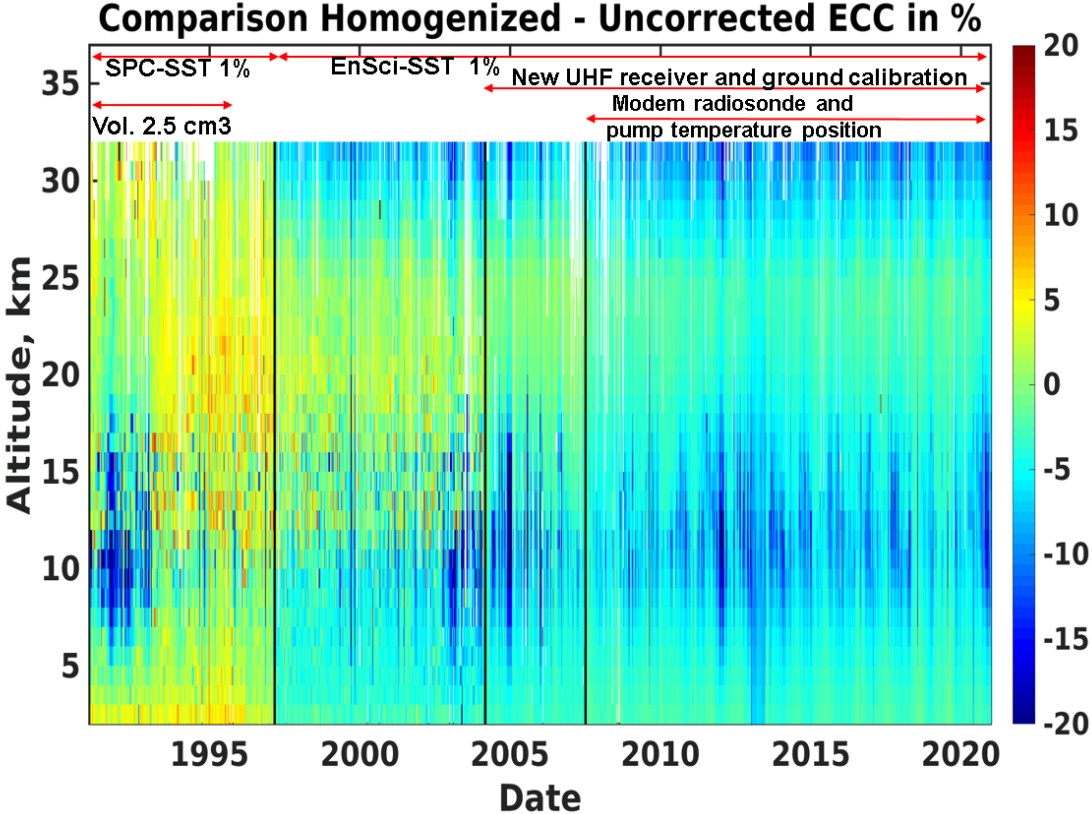

**Figure 2.** Time evolution of the relative difference between homogenized and uncorrected ozone concentrations as a function of altitude. Color scale is in %. Major changes in the sounding procedure or processing are shown in the top part of the figure. Pump flow rate corrections and removal of $I_b$ pressure dependency are applied to the entire data set.

The homogenization procedure also includes a retrieval of the uncertainty on the ozone partial pressure at each vertical level. The median value of the relative error on the ozone concentration measured by the ECC is of the order of 6-7% in the stratosphere and 7-9% in the troposphere (see section 4.2 showing the vertical distribution of the relative error of the ECC ozone concentrations used for the lidar-ECC comparisons).

**3  Data and homogenization assessment**

In this work, the benefit from homogenization of the ECC ozonesonde timeseries is assessed using comparison of homogenized and non homogenized ECC ozone concentrations with other ozone measurements carried out at OHP. First these comparisons can be done as a function of altitude using either Ultraviolet DIfferential Absorption Lidar (UV-DIAL) or Microwave Limb Sounder (MLS) satellite observations in the stratosphere (Froidevaux et al., 2008). Two UV-DIAL have been operated at OHP:





the first one, LiO3St, is optimized for stratospheric $O_3$ profiling between 10 and 50 km ASL using an absorbed wavelength at 308 nm and a reference wavelength at 355 nm (Godin-Beekmann et al., 2002) and the second one, LiO3Tr, is optimized for tropospheric ozone monitoring between 2.5 and 14 km ASL using the 289 nm and 316 nm wavelength pair (Ancellet and Beekmann, 1997). Regular nighttime measurements (2-4 per week) have been made with LiO3St since 1985 and with LiO3Tr since 1990. The LiO3Tr is known to be accurate in the 6 km to 10 km altitude range with smallest systematic lidar error ($<$

8%) due to overlap function mismatch between the two wavelengths at ranges below 4 km and to background signal correction due to photomultipliers (PMT) non linear response above 10 km (Ancellet and Ravetta, 2003). The LiO3St best accuracy ($\leq$5%) is generally in the 15-40 km altitude range when ozone concentrations are large enough to minimize lidar systematic errors and when signal-to-noise (SNR) ratio is maximum (Nair et al., 2011). The retrieved ozone concentration error at each vertical level for LiO3Tr takes into account the statistical error due to the detection noise and the systematic error due to the

background signal correction and to the residual error below 4 km when correcting the effect of a differential overlap function. The retrieved ozone concentration error for LiO3Str is calculated using the recommendations of Leblanc et al. (2016), not including a possible systematic error of 2% due to the $O_3$ absorption cross section accuracy. To minimize the impact of spatio-temporal variability of ozone concentrations on the analysis of lidar-ozonesonde comparisons (Liu et al., 2013), only nighttime soundings with a time lag of less than 2 hours or 6 hours were considered in the troposphere or the stratosphere, respectively.

We end up with a set of about 40 profiles for each lidar with minimum spatio-temporal differences between 1994 and 2021. The time distribution of the number of lidar profiles per year for optimal comparison with ECC is shown for LiO3Tr and LiO3St in Fig.3.

The 2005-2020 stratospheric profiles from AURA MLS v5 level 2 files have been also retrieved from 56.23 hPa to 6.81 hPa with a vertical resolution of the order of 2 km at these levels. The overpass criteria is $\pm 5$deg latitude, $\pm 8$deg longitude,

and all MLS profiles meeting this distance criteria within one day of the sonde are averaged to make the comparison with the ozonesonde. Although the spatio-temporal differences between ECC soundings and satellite overpasses will be greater using these criteria, we obtain many more comparisons than by restricting ourselves to nighttime soundings. The ECC/satellite comparison will then be complementary of the ECC/lidar difference analysis.

Secondly comparisons of total ozone column (TOC) are also useful to check the added value of the homogenization in the

130 stratosphere. Beside the TOC provided by the OHP spectrophotometer (Hendrick et al., 2011; Van Roozendael et al., 1998), satellite TOC level 2 overpass files by AURA Ozone Monitoring Instrument (OMI), Suomi-NPP Ozone Mapping and Profiler Suite (OMPS) and Global Ozone Monitoring Experiment (GOME) 2A and B have been selected when they are within 12 hours of the ozonesonde and picking the closest pixel in time and space to the ozonesonde station. The vast majority of L2 TOC are within 100 km of OHP. The corresponding ECC sounding TOC used for the satellite comparison, is calculated using

the integration of the ozone concentrations up to 10 hPa and the McPeters and Labow (2012) climatology above 10 hPa.

Thirdly the benefit of homogenization on long term ozone trends for several altitude ranges in the troposphere and the stratosphere, has been studied using all the lidar and ECC measurements made at OHP. The lidar monitoring period is indeed as long as the ozonesonde data set, and includes the major ozonesonde procedure or ozonesonde type changes in 1997, 2004 and 2007. Only simple linear trends of the ozone concentrations corrected for the mean seasonal variation at OHP will be



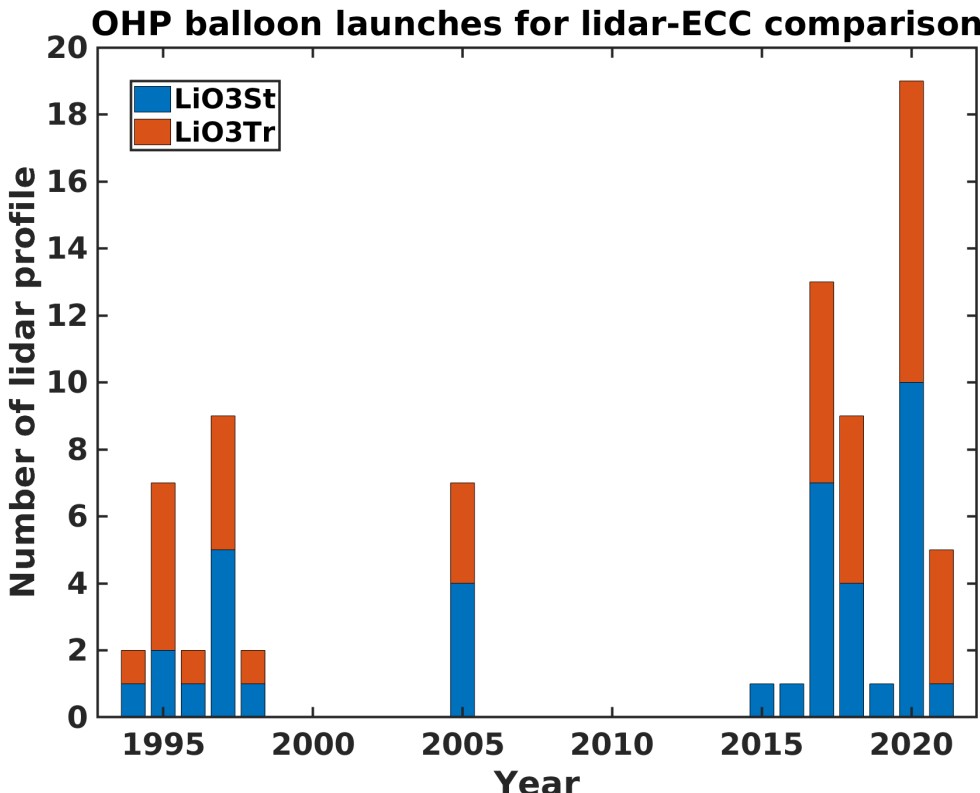

**Figure 3.** Time distribution of the number of OHP lidar observations per year with a time lag between ECC launches and lidar measurement < 2 hours (LiO3Tr) or < 6 hours (LiO3St). These observations are used for the lidar-ECC comparison of Fig.5 and 6. The blue bars correspond to LiO3St measurements and red bars to LiO3Tr.

considered in this study for the assessment of the homogenization. A more comprehensive trend analysis for the OHP would need either multiple linear regression model as described in Hassler et al. (2014) or Thompson et al. (2021) for the stratosphere or statistical regularization method as described in Chang et al. (2020) for the troposphere when data sampling is sparse. For the stratospheric trend the period 1990-1995 is removed to minimize interferences by the eruption of Mt Pinatubo. Lidar and satellite ozone observations cannot be retrieved in the lowest atmospheric layer below 2 km ASL, $O_3$ surface observations
are then included using measurements of a TEI-49C instrument with an air intake on the roof of the lidar building. Data are recorded continuously since December 1997, except between June 2010 and August 2012 when the responsibility for surface ozone measurements was entrusted to ATMOS-SUD air quality network at the same location. Data between July 2002 and July 2003 have been also removed because of a contamination problem in the air intake. The trend of $O_3$ surface observations is compared to ECC trends using either all the daily mean $O_3$ surface data available since 1998 and corrected for the mean
seasonal variation or the hourly mean $O_3$ concentration measured on the day and time of the ECC launch and also corrected





for the corresponding mean seasonal variation. A significant difference between the two surface concentration trends will point towards a strong sensitivity of the trend sign to the limited number of observations by the ECC.

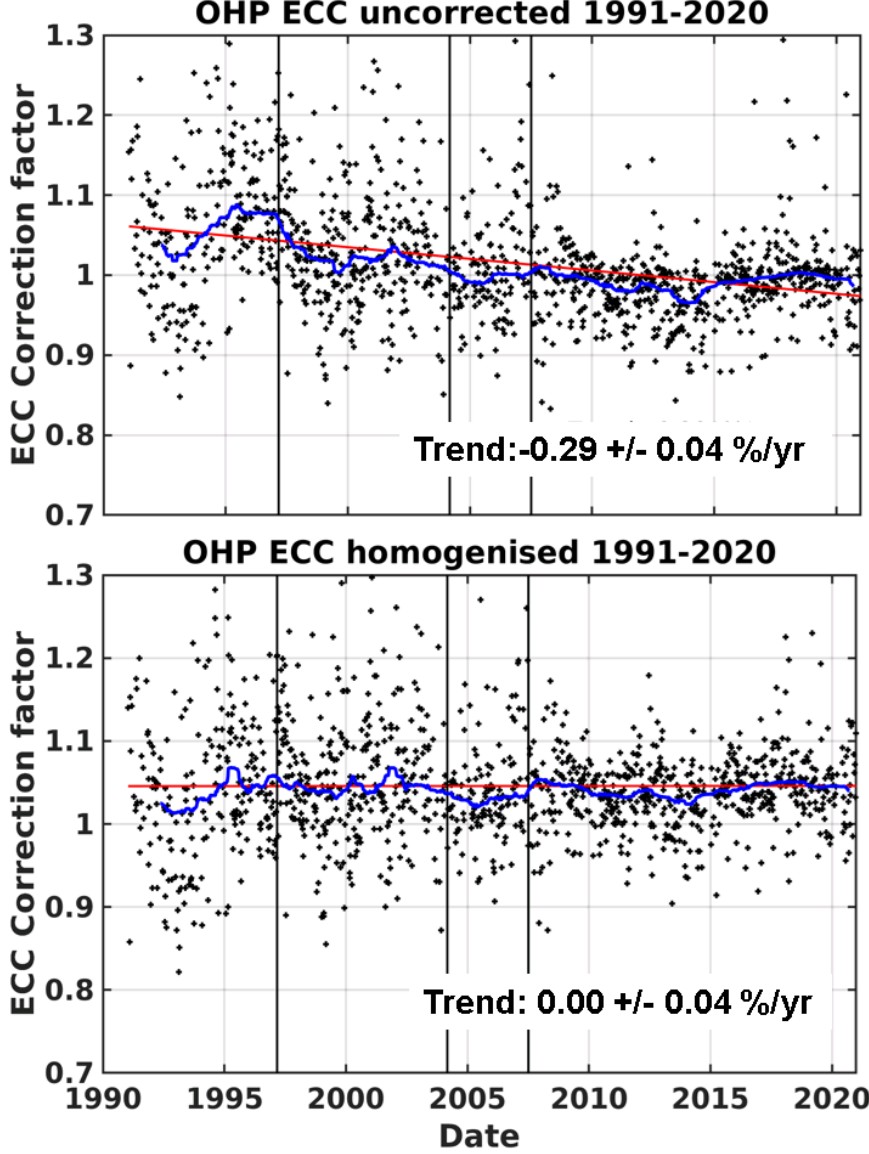

**Figure 4.** Time evolution of the OHP ECC normalization factor ($N_T$) from 1991 to 2021 before (top panel) and after homogenization (bottom panel). The black lines correspond to the major ozonesonde changes in 1997, 2004 and 2007 (see Fig.2). The thick blue lines are the 100 points, centered, moving averages. $N_T$ linear trends are shown in red, while the slope and its uncertainty with a 95% confidence are given in % per year.



## 4 Results and discussion

### 4.1 Normalization Factor Trend

The time evolution of the normalization factor $N_T$ is plotted in Fig. 4 for the uncorrected and homogenized OHP ECC sondes. The major changes in the ozonesonde supplier or the ozonesonde conditioning shown in Fig. 2 are also reported in Fig. 4. The uncorrected $N_T$ time evolution shows that the dispersion of points is greater before the switch to MODEM ozonesonde in 2007, but more concerning is the significant negative trend of -0.29±0.04% per year which is as large as reported trends in the troposphere (Gaudel et al., 2018). The homogenized $N_T$ does not exhibit a significant trend (0.00±0.04%/year) anymore,

showing the strong benefit of the homogenization. However the average normalization factor for the whole date record is not equal to 1 but shows a likely -5% bias ECC TOC compared to the OHP spectrophotometer. This may be partly due to the calculation of residual ozone above the burst altitude and partly to a possible bias in the stratosphere. The calculation of the residual ozone which represents 7-10% of the TOC depends strongly on the last ozone concentrations measured before the balloon burst. An underestimation of these concentrations of the order of 10% (e.g. due to freezing or evaporation of the

ozonesonde solution) would lead to an underestimation of the order of 2% of the TOC. A negative bias of about 3% in the stratosphere is still necessary to explain an average normalization factor of 1.05.

### 4.2 Nighttime Ozonesonde/Lidar comparison

The ozone concentration vertical profiles of ECC ozonesondes launched with a time lag less than 2 hours after or before a LiO3Tr lidar profile have been divided into six 1.5-km vertical layers between 3 and 12 km. The relative differences between

170 the ECC and lidar $O_3$ concentration are calculated for each 1.5 km vertical bin. The mean of the relative difference and its uncertainty are then calculated for the 40 profiles, the time distribution of which is shown in Fig.3. The uncertainty of the mean difference in a 1.5-km vertical interval for a single $O_3$ profile is based on mean absolute error of both lidar and ECC measurements (see section 2 and 3) at each recorded altitude in the corresponding 1.5-km vertical interval. The uncertainty of the overall mean difference is then retrieved assuming an independent error for the 40 comparisons taken into account. The

175 mean relative differences between the homogenized ECC and LiO3tr show a non-significant bias of the order of 1% for the altitude range 4.5 to 9 km considering the error on this difference which is of the order of ±2% (Fig. 5a). The mean relative differences between the uncorrected ECC concentration and LiO3tr however show a significant bias of the order of +4% in the same altitude range. It is due to the error when using uncorrected EnSci-SST 1% and a pressure dependent background current. The comparison between the altitude dependence of the error of the lidar measurement and that of the ECC measurement in

the troposphere (Fig. 5b) shows that the ECC error stay in the range 7%-9%, while the lidar is less accurate (error > 9%) below 4.5 km and above 11 km. Below 4 km the significant bias of -4% between the homogenized ECC and the LiO3tr can be then explained by the large uncertainty on the $O_3$ retrieval by the LiO3tr (> 10%) because of the sensitivity to the overlap function correction in this altitude range. Above 10 km the large difference between lidar and ECC (≈10%) is due to different spatial resolutions for the two measurements in a region with strong $O_3$ vertical gradients and due to the increasing error on the LiO3tr

measurements in the lowermost stratosphere.

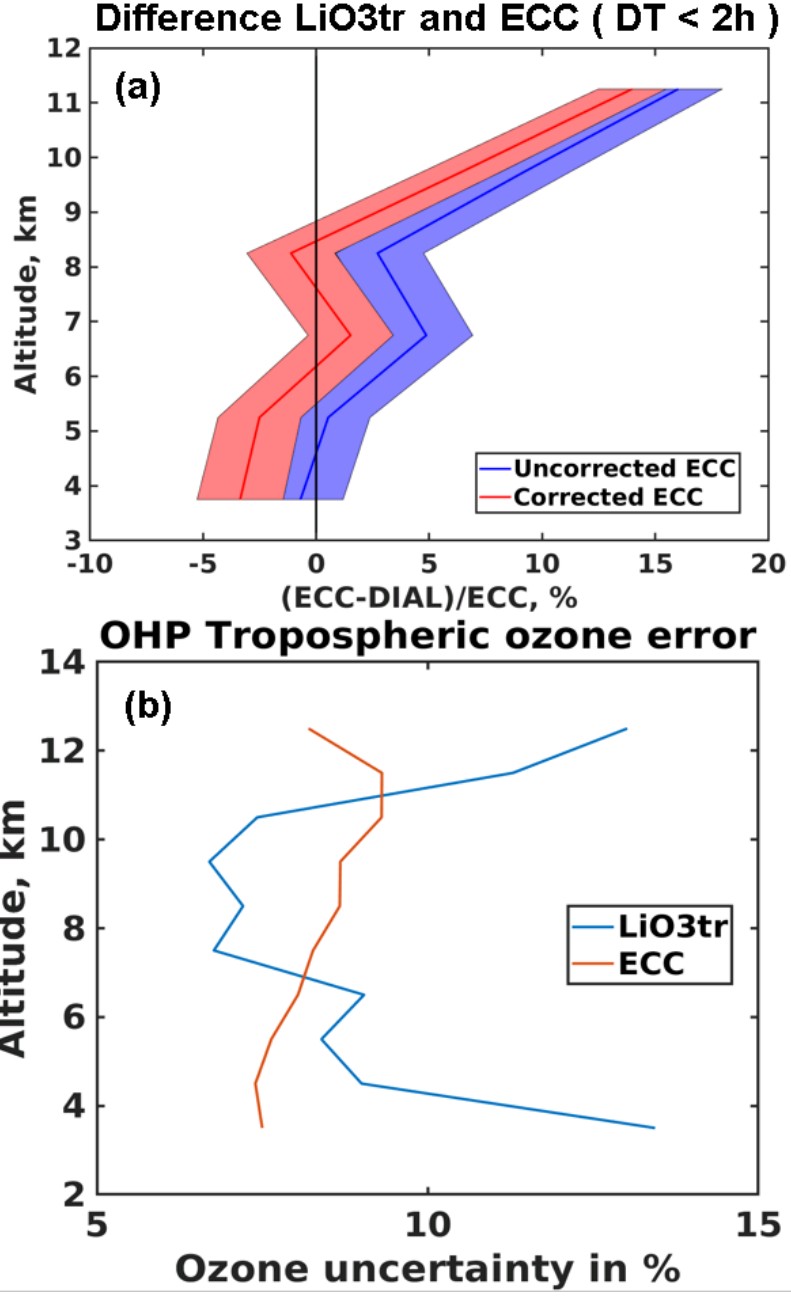

**Figure 5.** (a) Mean relative ECC - LiO3Tr ozone concentration differences in % between 3 and 12 km for uncorrected (blue) and homogenized (red) ozonesonde. Shaded areas represent the error on the mean difference. (b) Vertical profiles of the median of the relative ozone concentration error in the troposphere for the homogenized ECC (red) and LiO3Tr (blue).



For the comparison with LiO3St, the time lag between ECC launch and lidar profiling period does not exceed 6 hours. The number of profiles is again about 40 and the time distribution is shown in Fig.3. The means of the relative difference between ECC and LiO3St are then calculated for 8 vertical layers between 14 and 30 km. As for the previous comparison with LiO3tr, the uncertainty of the mean difference between the two instruments is retrieved assuming an independent error for the 40 comparisons taken into account. The mean relative differences between the homogenized ECC and LiO3St still show a significant bias of the order of -5% between the ECC and LiO3St above 17 km with an error on the mean difference which is of the order of $\pm$1.5% (Fig.6a). This difference becomes less than 1% near 15 km. In contrast to the LiO3Tr comparison, the difference between the homogenized and uncorrected ECC measurements is small ($\approx$2%), except above 28 km where the homogenized ECC concentrations are even lower than the lidar concentrations by -10% (Fig.6a). The comparison between the altitude dependence of the error of the lidar measurement and that of the ECC measurement in the stratosphere (Fig. 6b) shows that the ECC error stay in the range 5.5%-6.5%, while the lidar is very accurate (error <2%) between 18 km and 30 km.

Considering the accuracy of the lidar observations in the stratosphere, frequent freezing or evaporation of the solution during nighttime soundings may be an explanation for the lowest performances above 28 km of the ECC launched at OHP. The $O_3$ partial pressure error related to a pressure offset for the Vaisala RS80 period may be another reason for such a difference, but this error will be limited as it exists for only one third of the ECC sondes used for this comparison (25 MODEM and 15 RS80 radiosondes). When looking at differences above 26 km between homogenized ECC and LiO3St for the MODEM and the RS80 subsets separately, there is indeed a smaller bias of -4% to -11% for the RS80 than the one of -6% to -9% for the MODEM. We have also considered two subsets with ECC pump temperature $T_i$ at 30 km either higher or lower than 290 K. The relative difference between the homogenized ECC and LiO3St $O_3$ concentrations above 26 km does not exceed 4% for the high $T_i$ subset while it ranges from -7% to -9% for the low $T_i$ subset. More investigations are needed to conclude that freezing or evaporation of the solution is indeed the major uncertainty above 26 km.

The -5% difference between LiO3St and the homogenized ECC in the stratosphere even after homogenization is consistent with the mean normalization factor of 1.05 discussed in the previous section. Indeed the means of the relative difference between LiO3St and ECC are no longer significant below 28 km when the ECC concentrations are multiplied by the normalization factor (black curve in Fig. 6). Notice however that such a correction would not be recommended for the tropospheric measurements.

### 4.3 Comparison Ozonesonde/Satellite

The comparison of satellite and ECC measurements covers a period from 2005 to 2021. In contrast with the ECC lidar/comparison all the ECC sondes have been used to calculate the ozone differences between the satellite observations and the ECC measurements. The vertical profiles of the relative differences with MLS are shown in Fig. 7 in the stratosphere from 20 km (50 hPa) to 31 km (10 hPa). The ozonesonde data are first averaged to 100 m vertical resolution and then interpolated onto the MLS pressure levels. While many differences exceed 5% when the ECC are uncorrected especially between 2010 and 2015, it is no longer true with the homogenized ECC data where the differences stay within the $\pm$5% interval. Above 28 km (15 hPa) we find again a negative bias reaching -10% when the ECC are homogenized. The differences between 20 and 28 km for the



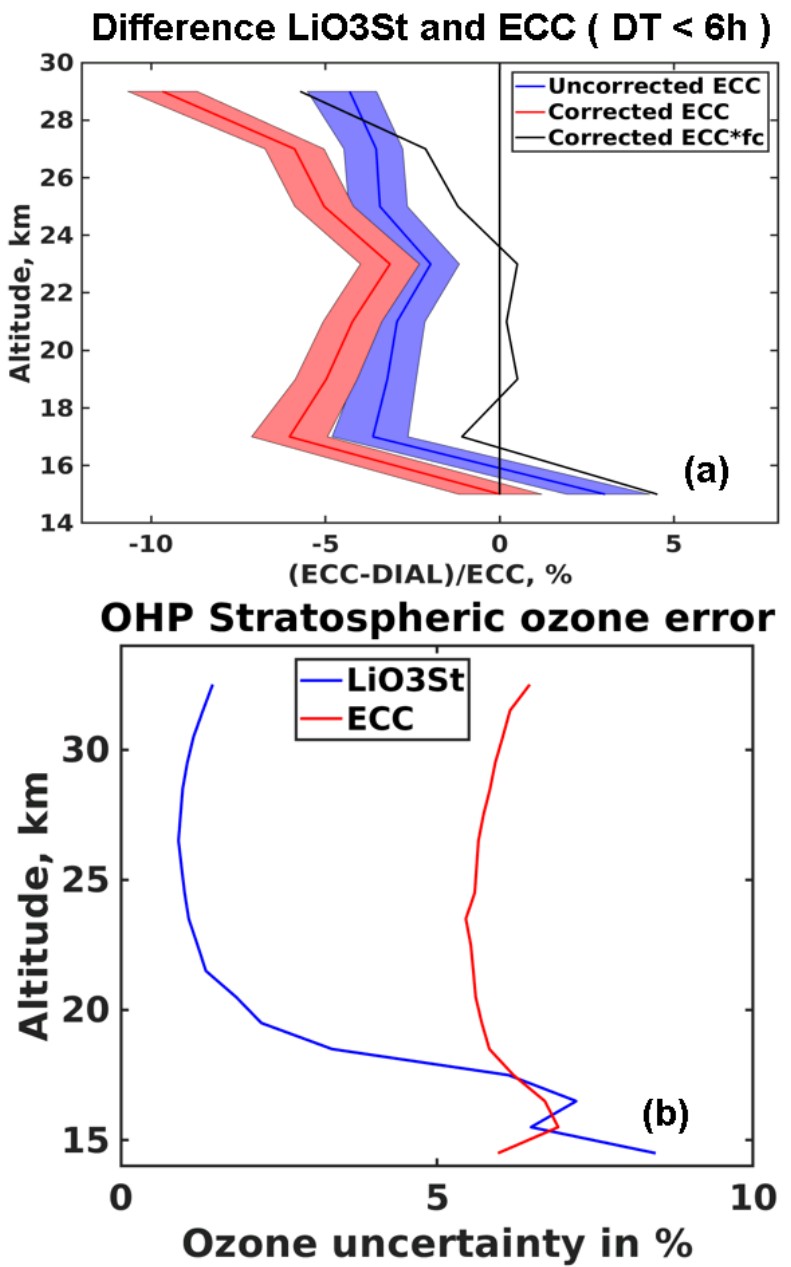

**Figure 6.** Mean relative ECC - LiO3St ozone concentration differences in % between 3 and 12 km for uncorrected (blue) and homogenized (red) ozonesonde. Shaded areas represent the error on the mean difference. The solid black line shows the mean relative differences when the homogenized ECC concentration is multiplied by the normalization factor $N_T$ (b) Vertical profiles of the median of the relative ozone concentration error in the troposphere for the homogenized ECC (red) and LiO3St (blue).





time period corresponding to the lidar/ECC comparison of section 4.2 (July 2017, March 2018, after May 2020) are within

the ±5% interval. The ECC/MLS comparison is then consistent with the results obtained in section 4.2 with the comparison

ECC/LiO3St. An interesting feature of this MLS/ECC comparison is the interannual variability of the differences and we see

that differences using homogenized ECC are more evenly distributed around zero.

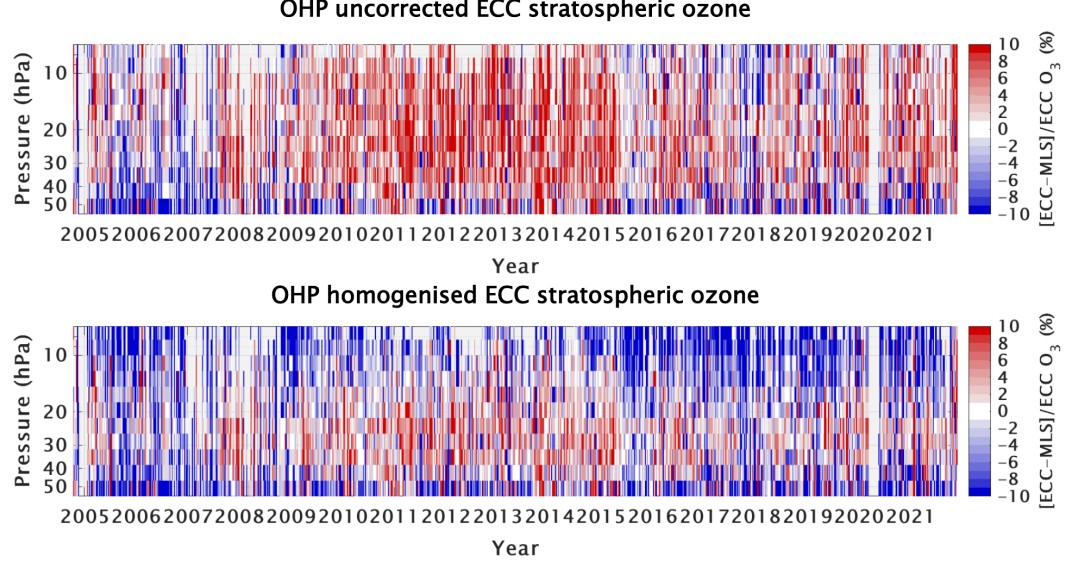

**Figure 7.** Relative $O_3$ concentration differences in % between ECC and MLS as a function pressure between 50 and 7 hPa before (top) and after (bottom) homogenization

    The time distribution of differences between the ECC TOC and the satellite TOC are shown in Fig. 8. No filtering for clouds

or distance is applied to lead to more available comparisons. The 100 point centered moving averages are superimposed on the

set of points corresponding to each single comparison. As expected the results are consistent with comparison between ECC and

stratospheric MLS profiles with the largest positive relative differences between uncorrected ECC and satellite TOC between

2010 and 2015. No post 2013 drop off in TOC measurement by the ECC is seen at OHP as observed at other measurement

sites in Stauffer et al. (2020). The OMI/OMPS and uncorrected ECC biases range between -1% to +5% while it is between 0

and +3% for GOME. The differences are mostly negative and between -4% and 1% after homogenization. So the satellite/ECC

TOC comparison is consistent with the time distribution of the normalization factor shown in Fig.4.

## 4.4    Comparison of trend analysis

### 4.4.1    Surface trend

First the inter-annual variation of homogenized and uncorrected ECC $O_3$ concentrations have been retrieved in the lowermost

troposphere (200-m layer above ground level) using the average ECC ozone concentrations in this layer. Such an inter-annual





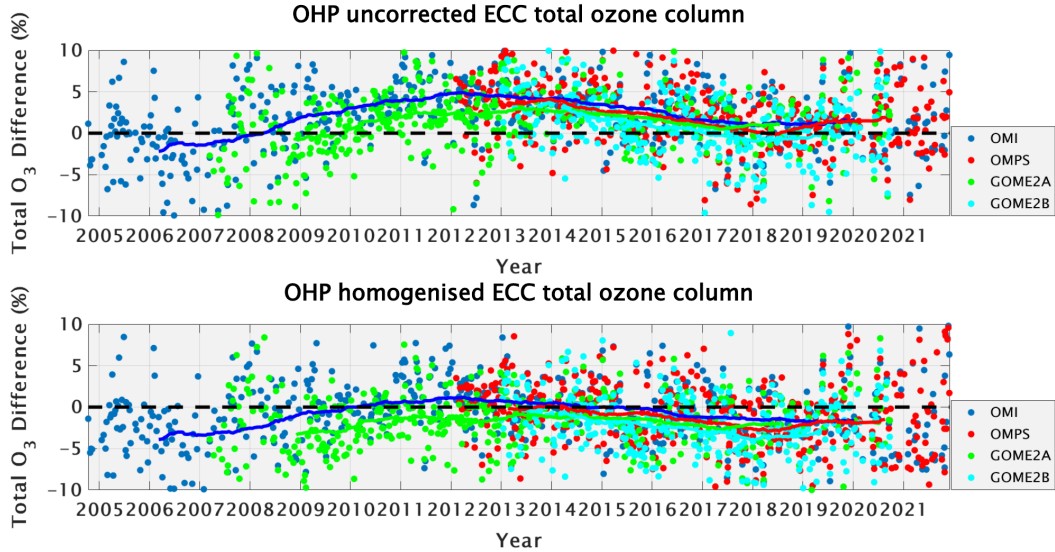

**Figure 8.** Relative differences between satellite and ECC total ozone column before (top) and after (bottom) homogenization. The thick lines are 100 points, centred, moving averages. No moving averages are plotted for less than 100 points.

variation can be compared with the one deduced from the surface ozone measurements made since 1998. To quantify the 22-years trend of ozone mixing ratio associated with this inter-annual variation, the latter is deseasonalised by subtracting from the surface mixing ratios the monthly averages calculated over the 22 years of data. This removes a major source of $O_3$ mixing ratio intra-annual variability which is of the order of 20 ppbv. The trends of the ozone mixing ratio and their 95%

confidence interval estimates are calculated using the regression lines across all the available deseasonalised mixing ratios. A weak negative trend of the order of -1.3±0.9 ppbv/decade is obtained for the uncorrected ECC deseasonalised mixing ratio (called ozone anomalies hereafter) and this trend changes very little (-1.1±0.7 ppbv/decade) after homogenization of the ECC (Fig.9). The ECC negative ozone trends compare very well with those obtained from surface measurements using either all the $O_3$ daily means between 1998 and 2021 (-1.3±0.2 ppbv/decade) or only the hourly means for ECC launching times (-1.1±0.6

245 ppbv/decade). The small difference between the trend calculated for all the surface daily means available and the trend using only the ECC launching times shows that the sensitivity of the trend magnitude to the sampling by ECC is not so large. The negligible difference between the uncorrected ECC trend and the homogenized ECC trend near the surface is mainly due to the fact that the same kind of correction are applied for all the data of 1998-2021 period, namely the scaling of EnSci-SST 1% response to the SPC-SST 1% one, the pump flow rate correction and the removal of $I_b$ pressure dependency. The difference of

250 13 ppbv between the 2003 positive yearly average of ozone anomalies and the 2005-2008 negative yearly anomalies for the uncorrected ECC (Fig.9a) is however slightly reduced to 9 ppbv for the homogenized ECC (Fig.9b). The difference between the positive 2003 yearly average of ozone anomalies and the negative one for 2008 does not exceed 8 ppbv for the surface measurements (Fig.9d), therefore in better agreement with the homogenized ECC inter-annual variations.

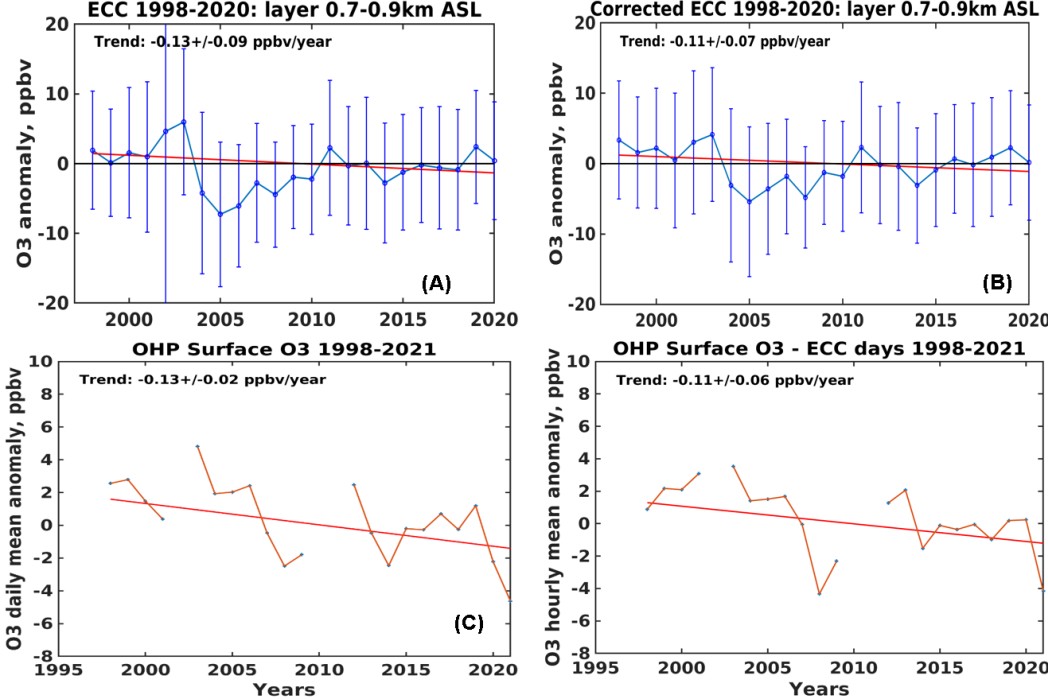

**Figure 9.** Interannual variation of deseasonalised $O_3$ mixing ratios in ppbv for the uncorrected ECC (A), homogenized ECC (B) and daily mean observations of the OHP surface $O_3$ analyser (bottom). The ECC mixing ratio are averaged in the 0.2-km layer above the surface. The surface daily mean observations are taken everyday (C) or for ECC launching day only (D). Blue circles are the annual medians of the deseasonalised $O_3$ anomalies with their standard deviation, while the red line represents the regression line through all the $O_3$ anomalies available between 1991 and 2020. Ozone trends in ppbv/year and their uncertainties with a 95% confidence are shown in each panel.

### 4.4.2 Tropospheric trend

Second the inter-annual variation of homogenized and uncorrected ECC ozone are compared in the free troposphere for three layers of 2-km thickness at 5 km, 7 km and just below the dynamical tropopause taken at 2 PV units (Fig.10). The three layers were selected in order to compare the ozone trends of the ECC sondes with those of the LiO3tr lidar. The mean altitude $Z_{tp}$ of the dynamical tropopause is 10.5 km at OHP (10 km in winter and 11.5 km in summer), so the upper layer approximately corresponds to the 8 km to 10 km altitude range. As for the surface trend retrieval, the mean ozone concentrations of the

layers are deseasonalised before calculating the trends of mixing ratio from the regression lines across all the 2-km ozone mixing ratio averages available in the 30-year database. The uncorrected ECC trends are always positive and significant and they increase with altitude, with the largest value (4.4±0.8 ppbv/decade) in the layer below the tropopause (Tab.1). The lidar also show significant positive trends for the 3 layers but with smaller values, e.g. 3.1±0.9 ppbv/decade below the tropopause. The lidar trends are in better agreement with the trends calculated using the homogenized values, e.g. 3.2±0.8 ppbv/decade

below the tropopause. Although the lidar and homogenized ECC yearly average of ozone anomalies do not overlap every year





considering the sampling differences, the main decennial changes are seen by both instruments above 6 km, namely the sign change of the anomalies between the period 2000-2010 and 2010-2020 (Fig.10). Overall the homogenization greatly improved the tropospheric trend assessment with smaller and more realistic values.

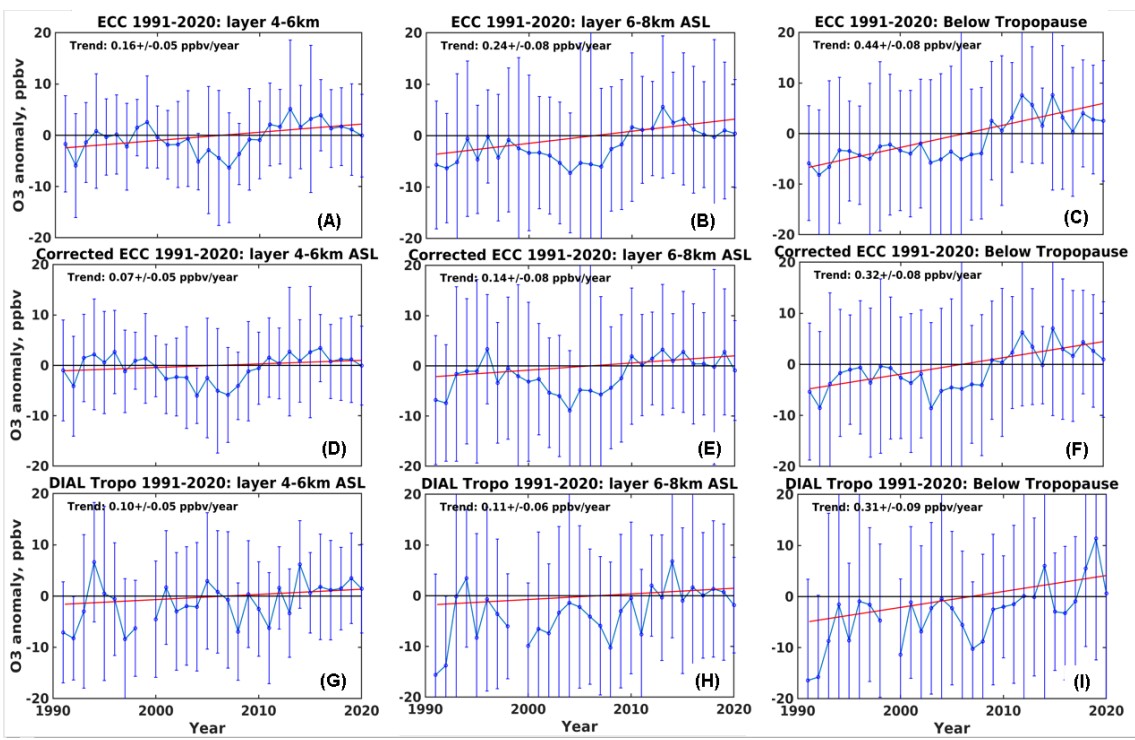

**Figure 10.** Interannual variation of deseasonalised O₃ mixing ratios in ppbv for the uncorrected ECC (top row), homogenized ECC (middle row) and LidO3Tr (bottom row) and 3 altitude ranges in the troposphere: 4-6 km (left column), 6-8 km (middle column) and the 2-km range below the 2 PVu dynamical tropopause (right column). Blue circles are the annual medians of the deseasonalised O₃ anomalies with their standard deviation, while the red line represents the regression line through all the O₃ anomalies between 1991 and 2020. Ozone trends in ppbv/year and their uncertainties with a 95% confidence are shown in each panel.

### 4.4.3 Stratospheric trend

Third the interannual variation of homonogenized and uncorrected ECC ozone are compared in the stratosphere for three layers of 2-km thickness at 19 km, 25 km and 29 km (Fig.11). The three layers were selected to be able to compare the ozone trends of the ECC sondes with those of the LiO3St lidar. The methodology developed for the surface and tropospheric ozone trends has been applied on the ozone concentrations given in molecules.cm⁻³ which is the primary unit used by the LiO3St for the ozone retrieval (Leblanc et al., 2016). The uncorrected ECC trends are always positive and significant ranging from
1.8±0.4 mol.cm⁻³/decade to 0.7±0.2 mol.cm⁻³/decade, while the trends retrieved from the lidar observations are negligible and not significant within the range -0.3±0.4 mol.cm⁻³/decade at 19 km to 0.2±0.2 mol.cm⁻³/decade at 29 km (Tab.2).





**Table 1.** Trend of lidar, uncorrected ECC and homogenized ECC in the troposphere. Last column corresponds to the 2-km layer just below the altitude of the dynamical tropopause ($Z_{tp}$)

| Altitude range | 4 to 6 km | 6 to 8 km | $Z_{tp}$ - 2 km to $Z_{tp}$ |
|---|---|---|---|
| LiO3Tr ppbv/yr | 0.10±0.05 | 0.11±0.06 | 0.31±0.09 |
| Uncorrected ECC, ppbv/yr | 0.16±0.05 | 0.24±0.08 | 0.44±0.08 |
| Homogenized ECC, ppbv/yr | 0.07±0.05 | 0.14±0.08 | 0.32±0.08 |

**Table 2.** Trend of lidar, uncorrected ECC and homogenized ECC in the stratosphere.

| Altitude range | 18 to 20 km | 24 to 26 km | 28 to 30 km |
|---|---|---|---|
| LiO3St mol.cm$^{-3}$/yr | -0.03±0.04 | 0.01±0.01 | 0.02±0.02 |
| Uncorrected ECC, mol.cm$^{-3}$/yr | 0.18±0.04 | 0.13±0.02 | 0.07±0.02 |
| Homogenized ECC, mol.cm$^{-3}$/yr | 0.07±0.04 | 0.04±0.02 | 0.03±0.02 |

The ECC trends using the homogenized ECC becomes also very weak within the range 0.7±0.4 mol.cm$^{-3}$/decade at 19 km to 0.3±0.2 mol.cm$^{-3}$/decade at 29 km. Although the trends are similar, the year-to-year variation of the homogenized ECC yearly average of ozone anomalies are generally smaller than the corresponding lidar yearly average at 19 km and 29 km (Fig.11). Such differences in the range of the yearly ozone anomalies are related to a different sampling for ECC and lidar profiling. The homogenization nevertheless greatly improved the stratospheric 30-years trend assessment with better agreement with the lidar analysis recognized as very accurate in the stratosphere (Nair et al., 2011).

## 5   Conclusions

The 30-years ozone data set from weekly ECC ozone soundings has been homogenized according to the recommendations of Smit et al. (2012). The major changes are related to the change of ECC sensor in 1997 (SPC-SST 1% to EnSci-SST 1%), the background current and internal sonde temperature corrections. The assessment of the OHP ECC homogenization benefit has been carried out using comparisons with ground based instruments located at the same station (lidar, surface measurements) and collocated satellite observations (MLS in the stratosphere and GOME/OMI/OMPS for the Total Ozone Column (TOC)). The major findings are:





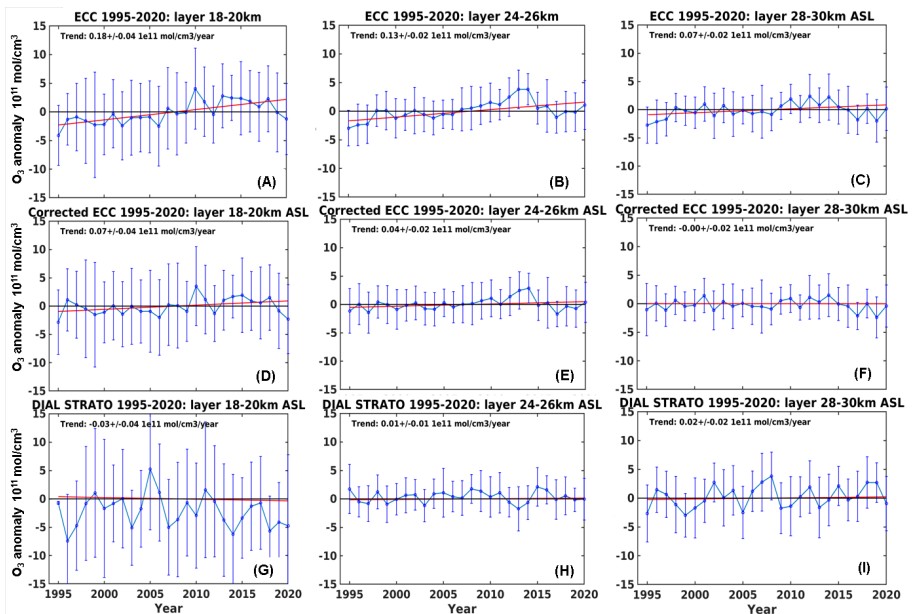

**Figure 11.** Interannual variation of deseasonalised $O_3$ concentrations in molecules.$cm^{-3}$ for the uncorrected ECC (top row), homogenized ECC (middle row) and LidO3St (bottom row) and 3 altitude ranges in the stratosphere:18-20 km (left column), 24-26 km (middle column) and 28-30 km (right column). Blue circles are the annual medians of the deseasonalised $O_3$ anomalies with their standard deviation, while the red line represents the regression line through all the $O_3$ anomalies between 1991 and 2020. Ozone trends in molecules.$cm^{-3}$/year and their uncertainties with a 95% confidence are shown in each panel.

- The 3-4 ppb positive bias of the ECC in the troposphere due to the use of uncorrected ENSCI-SST 1% and a pressure dependent $I_b$ is corrected with the homogenization leading to a better agreement between the LiO3Tr lidar and ECC in the mid-troposphere

- The ECC trends of the seasonally adjusted ozone concentrations are significantly improved both in the troposphere and the stratosphere when the ECC concentrations are homogenized, as shown by the ECC/lidar or ECC/TEI trend comparisons.

- The negative trend of the normalization factor ($N_T$) calculated using the OHP SAOZ total column disappears thanks to the homogenization of the ECC. There is however a remaining -5% negative bias which is likely related to an underestimate of the ECC concentrations in the stratosphere above 50 hPa as shown by comparison with the OHP LiO3St lidar and MLS (no bias between ECC and lidar when the ECC is multiplied by $N_T$). The reason for this bias is still unclear and must be better understood.

- Differences between TOC measured by ECC and by GOME or OMI/OMPS switch from 2%±2% for uncorrected ECC to -2%±2% for homogenized ECC being consistent with the $N_T$ time distribution.



- Direct comparisons of homogenized and uncorrected ECC concentrations in the stratosphere between 18 km and 30 km only show limited changes using a subset of 40 days with LiO3St and ECC measurements time difference less than 6 hours, but differences between MLS and ECC using all ozonesondes from 2005 to 2021 are more evenly distributed around zero for the homogenized time series than for the uncorrected ECC.

- Both the comparisons with lidar and satellite observations suggest that homogenization increases the negative bias of the ECC up to 10% above 28 km

*Code and data availability.* OHP ECC data are available at https://doi.org/10.25326/293. LiO3St and LiO3Tr data are available at the NDACC web site (https://www-air.larc.nasa.gov/missions/ndacc/data.html?station=haute.provence/ames/lidar/) MLS/Aura Level 2 Ozone (O3) Mixing Ratio V005 are available at https://doi.org/10.5067/Aura/MLS/DATA2516. OMI/Aura Ozone (O3) Total Column Daily L2 are available at https://doi.org/10.5067/Aura/OMI/DATA2025. OMPS-NPP L2 NM Ozone (O3) Total Column L2 are available at https://doi.org/10.5067/0WF4HAAZ0VHK. GOME 2A and B are available at http://www.eumetsat.int. Meteorological Analysis are available at ECMWF (http://www.ecmwf.int). The OHP ECC homogenization code is available on request (Renaud Bodichon <rboipsl@ipsl.jussieu.fr>).

*Author contributions.* G. Ancellet, S. Godin-Beekmann and H. Smit proposed the study. HS and R. Van Marderen provided the guidelines for the homogenization of the OHP ECC. R. Bodichon wrote the homogenization software for the OHP ECC. GA, SGB and A. Pazmino provided the OHP lidar and SAOZ data. GA carried out the analysis and wrote the paper. R. Stauffer carried out the comparison of ECC and satellite data. All the authors contributed to discussion and feedback essential to the study.

*Competing interests.* No competing interests are present

*Acknowledgements.* The work was supported by CNRS-INSU and ACTRIS through the project NDACC-France. The technical staff of Observatoire de Haute Provence is gratefully acknowledged for carrying out the lidar measurements and ECC ozone soundings. We acknowledge the European Centre for Medium Range Weather Forecasts (ECMWF) for providing meteorological reanalysis for lidar data processing and dynamical tropopause altitude calculation. The authors acknowledge NASA/GSFC for providing the satellite data used in this paper (MLS, GOME, OMI)



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
