# Peer review of "Homogenization of the Observatoire de Haute Provence ECC ozonesonde data record: comparison with lidar and satellite observations"

_Atmospheric Measurement Techniques, 2022_

## Author Response (AR1)

**Response to Reviewer 1**

**This document is the list of our responses to the reviewer's comments and a revised version of the text is also attached to this response to show the changes in red and the deleted sentences using strikethrough text**

This is one of the first papers describing the results of homogenizing a historic ozone sonde record by applying the corrections suggested by the OzoneSonde Data Quality Activity (Smit et al., 2012; Smit and Thompson, 2021). The paper is generally well written and scientifically solid. The English could benefit significantly from copy editing. My major suggestions are to improve the Figures, and add additional important information. After that, the paper is well suited for publication in AMT.

**We warmly thank the reviewer for his/her suggestions and comments. We have modified figures 6 and 9 to 11 to add new data requested by the reviewer or to facilitate the comparison of the different information initially provided. A careful copy editing of English writing has been made.**

Major suggestions:

The tight < 6 hours coincidence criterium for matching sonde ascents and lidar data results in only about 40 matches (sondes launched at night, ~4% of all sondes), out of more than thousand sondes (most of which are launched during daytime). I suggest to also allow < 12 hours coincidence, which will match many more sondes with the nighttime lidar measurements. It would be very interesting to see if this changes the results and statistics presented in Figs. 5 and 6.

**We agree that above 20 km the comparison of lidar and ECC measurements is still relevant with a time shift of 12h instead of the 6 h initially used. Such a comparison has been added in Fig. 6 for the period corresponding to the comparison with the MLS measurements. The number of coincidences increases from 40 to 366. Although the difference between uncorrected and homogenized ECC is now very significant, the bias between LIO3St lidar measurements and homogenized ECC remains of the same order of magnitude at altitudes greater than 20 km. The following sentence has been also added in section 3. Line 137:** « For the sake of a more complete discussion of the two types of comparisons made in the stratosphere, we also considered a lidar data set of 366 profiles from 2005 to 2021 with less restrictive measurement time difference with the ECC launches (<12h). Such a criterion is valid as long as the rapid O3 variations typically encountered below 18 km are not included. »

Are geopotential or geometric altitudes used in the sonde vs lidar comparison? This needs to be clarified. If geopotential altitudes are now used for the sondes, switching to geometric altitudes (as used by the lidar) should improve the comparison above about 25 km.

**We thank the reviewer for this insightful suggestion. This decreases differences between the ECC and LiO3St or MLS above 26 km (from -10% to -8%). All comparisons shown in Figure 6 have been recalculated with geometric altitude. The following sentence has been added in section 4.2 line 208:** "The means of the relative difference between ECC and LiO3St are then calculated for 8 vertical layers between 14 and 30 km using the geometric altitude for the ECC sondes as geopotential altitudes become significantly greater than lidar geometrical altitudes above 25 km"

In addition to showing the mean differences sonde minus lidars in Figs. 5 and 6, the authors should also show the mean difference sonde minus MLS (R. Stauffer is one of the co-authors, and should be able to provide that quite easily). This would be important to compare with the stratospheric sonde-lidar difference. It might help to elucidate the significant ~5% difference seen in the stratosphere. From Fig. 7 it looks like the ECChomogenization is moving the sondes in the right direction, and that there is no significant difference between homogenized sondes and MLS around 20 hPa (~26 km). This is different from Fig. 6, where the homogenization seems to move the sondes in the wrong direction.

**We agree and ECC-MLS comparisons are now also shown in Fig. 6b. Indeed the ECC-MLS and ECC-LiO3St do not have the same sign after the homogenization.The following paragraphs are now included in section 4.2 line 217:**

« For the period 2005-2021 and using a time difference less than 12 hours, the negative bias between the homogenized ECC and the lidar decreases down to -2% between 22 and 24 km, but remains as large as -7% above 28 km (Fig.6b). Note also that the mean uncorrected ECC and lidar difference is now slightly positive (+1%) for the 2005-2021 period in good agreement with the $N_T$ negative trend shown in Fig.4. Below 18 km, the -4% negative bias between homogenized ECC and lidar (Fig.6b) should be interpreted by possible significant concentration changes within 12 hours in this altitude range. »

**and section 4.3 line 258**

« The fact that the average ECC-MLS difference shown in Fig. 6b is slightly positive (+2%) in the 22-26 km altitude range, while the average ECC-LiO3St difference is slightly negative (-2%) means that homogenization is a good compromise for intercomparability with other techniques measuring O3 in the stratosphere below 26 km. Above 26 km, both comparisons indicate a negative bias in homogenized ECC O3 concentrations of less than -6% »

Figures 5 and 6 should be combined into one Figure. The sonde - MLS differences could be included in that Figure as well

**The instruments used to plot figures 5 and 6 respectively in the troposphere and stratosphere are very different and the impact of ECC homogenization is not the same for these two regions. We therefore prefer to keep 2 separate figures for these 2 discussions.**

Figure 9 needs to be improved. The different axes make comparison of ECC and surface data difficult. It might be better to inlcude the surface data directly in the ECC

**Figure 9 (now 10) is modified in order to be able to superimpose on the same temporal evolution of the annual averages both the measurements of the ECC and those of the OHP surface O3 analyzer.**

Figure 10 should also show the MLS time series. That would be very helpful.

**The MLS data record only starts in 2005 and is not as relevant as the lidar data record starting in 1990 for the evaluation of homogenization on the sign of the O3 trends or its magnitude in the stratosphere. The differences between uncorrected and homogenized ECC are not large enough after 2005 to significantly alter the long-term trend in O3. We believe that only the analysis of the temporal evolution of the difference between ECC and MLS, shown in Figure 8, is relevant to this paper.**

Figure10 it is not necessary to plot all the error bars, since they are all very similar. Instead, I think it would be much better to show all respective time series (ECC old, ECC homgenized, Lidar, MLS) in one plot.

**We agree with the reviewer. The annual standard deviation does not change a lot and all the time series can be shown in a single plot for a given altitude range. Fig. 10 and 11 (now 11 and 12) have been modified accordingly (not including MLS as explained above).**

While the overall trends and their comparison is useful, it would also be quite important to look at time series of ECC minus surface, ECC minus lidar(s), and ECC minus MLS. Are there significant trends in these difference time series? Is there a significant annual cycle in these differences? How do trends (and possibly remaining annual cycles) change with the homogenized ECC data? These difference time series probably do not require subtraction of an annual cycle. Since common variations largely cancel out, trend uncertainty should be smaller than when comparing trends of the individual monthly or annual anomalies.

**We understand the reviewer's point of view. Such an analysis is complementary to, but different from, the trend analysis presented in Figures 11 and 12. The purpose of our trend analysis was to**

use all weekly ECC observations to assess how sensitive is a linear ozone trend to homogenization of the ECC. For our study the temporal evolution of the differences between two instruments is only relevant when using observations made on the same day. Such a discussion was already present in section 4.3 for the MLS measurements shown in Fig. 7 (now 8) because the time lag with the ECC measurements is less than 1 day and there are many MLS occurrences. This is not always the case for comparisons with lidar (only 366 profiles out of 1412). Nevertheless, the temporal evolution of the difference between ECC and lidar between 2005 and 2021 is now included in Figs. 6c,d and this temporal evolution is compared with the MLS time series already discussed in section 4.3. The following paragraphs are now included in Section 4.2 line 222:

« The time evolution of the relative difference of ECC and LiO3st ozone concentrations is shown in Fig 6c and Fig. 6d for uncorrected and homogenized ECC, respectively. Many of the differences between uncorrected ECC and LiO3St are greater than +6% between 2007 and 2016 while there are some negative differences approaching -6% in 2006. Homogenization improves the relative differences now remaining between -5% and +5\%, except in 2006 when the negative bias decreases down to values smaller than -6%. »

and in section 4.3 line 255

« An interesting feature of this MLS/ECC comparison is the interannual variability of the differences. It can be observed that differences using homogenized ECC data are more evenly distributed around zero. The same conclusion could be drawn from the time evolution of the relative differences between homogenized ECC and LiO3St presented in Fig. 6c,d. »

For all trend uncertainties: Is autocorrelation of the residuals accounted for? Please state that, and preferably account for it.

As explained in section 3, only basic linear trends of the ozone concentrations corrected for the mean seasonal variation at OHP is considered in this study for the assessment of the homogenization. The following sentence as been added in section 3, line 153 : « The trend uncertainties are calculated using the 95% confidence limit of the slope of the linear regression assuming that the residuals are not correlated for weekly (ECC) or 2/3 per week (lidar) observations). »

Regarding the average differences between corrected / uncorrected OHP ECCs and satellite total ozone, as well as MLS ozone profiles: It would be important to compare the OHP results / biases with those seen at other sonde stations. Since R. Stauffer is a coauthor, and has most of these data, an additional paragraph, or even additional Figures would be very important. This is needed to put the OHP results into the necessary wider context.

The purpose of this work is to focus on the reanalysis of OHP because we can use both lidar and satellite observations for this site. Furthermore, the corrections are very site-dependent and the conclusions drawn from the use of O3S-DQA at OHP are not easily applicable to other sites.

Nevertheless, the following paragraph has been added in the conclusion line 359 to compare our results with the homogenization performed in the SHADOZ network and for the Uccle/De Bilt sites in Europe : « While the objective of this paper is to discuss the impact of homogenization on the OHP dataset using lidar and satellite measurements, it is worth checking how such corrections have improved data quality at other sites. The impact of the homogenization is dependent on the site, because different homogenization steps have to be applied at different stations. In general, the additional corrections for the pump temperature will give higher ozone partial pressure amounts in the stratosphere. On the other hand, applying a constant background current subtraction instead of a pressure dependent background current and applying the transfer functions from EnSci-SST 1% will lead to lower ozone partial pressure values above 10 km. Witte et al. (2017) performed an extensive analysis of 7 SHADOZ network stations in the tropics, showing that the mean differences between ECC and MLS are reduced from -11.2±13.6% to -3.0 ±10% at 40

hPa (22 km) and from -3.2%±4% to -0.7±3.1% at 17 hPa (28 km). In Europe, Van Malderen et al. (2016) observed that the O3S-DQA corrections actually give higher (+1%) and lower (-2%) ozone concentrations in the stratosphere with respect to standard processing for the Uccle 1997-2014 and De Bilt 1993-2014 ECC observations, respectively. This is mainly due to the fact that the pump temperature correction was a major correction for Uccle, while changing the background current correction has a major effect for De Bilt. O3S-DQA corrections reduce the relative $O_3$ difference between Uccle and De Bilt in the lower stratosphere. The analysis of homogenized ECC at OHP using LiO3St or MLS show similar improvements in the stratosphere below 26 km. The remaining bias of -2% to -3.7% between homogenized ECC and other techniques measuring $O_3$ in the stratosphere at OHP is also in the range of the remaining negative differences between homogenized ECC and MLS observed in the 22 to 28 km altitude range by 4 stations of the SHADOZ network (Witte et al., 2017).}"

**Response to Reviewer 2**

**This document describes our responses to reviewer's comments and a revised version of the text attached to this response shows the new figures, the text changes in red and the deleted sentences using strikethrough text**

*This is an excellent paper, as I would expect from such an expert group of authors. It is a welcome addition to the global effort to re-evaluate ozonesonde records, and certainly appropriate for publication in AMT. I have just a few points that the authors should address before publication.*

**We warmly thank the reviewer for his/her suggestions and comments. We have modified figures 6 and 9 to 11 to add new data for a better characterization of the bias between homogneized ECC and other measurement techniques in the stratosphere. A careful copy editing of English writing has been made.**

*My major issue is with the confusing way that uncertainty is discussed, and often carelessly referred to as "error". For example in Section 3, 4.2 and Figures 5 & 6. The term "error" seems to be used interchangeably with "uncertainty". They do not mean the same thing, and in most cases where "error" is used, I think the authors mean "uncertainty". Since error is the difference between the measurement and the true value, No doubt the authors in fact mean the ECC (random? systematic? overall?) uncertainty. This is made worse by Figures 5b and 6b, where the captions to 5a and 6a state "Shaded areas represent the error on the mean difference." The text calls this an uncertainty. I expect the shaded areas should be described as confidence intervals, and it should be stated whether they show one standard error (more properly standard uncertainty) of the mean, or two (the latter being a conventional 95% confidence interval). If in fact they are standard deviations, then that should be stated.*

**We apologize for the confusion due to the inappropriate use of the word error in several sentences. We agree with the reviewer that error should be often replaced either by uncertainty in section 3 and 4.2 or confidence limit in section 4.4. It was corrected in several sentences. Captions of figure 5 and 6 have been also corrected. Description of the shaded area in Fig. 5 and 6. are now described by the following paragraph in section 4.2 line 188:**

"The uncertainty of the mean difference in a 1.5-km vertical interval for a single $O_3$ profile is based on mean absolute uncertainties (systematic and statistical) of both lidar and ECC measurements (see section 2 and 3) at each recorded altitude in the corresponding 1.5-km vertical interval. The statistical standard uncertainty of the overall mean difference is then retrieved assuming that the 40 comparisons are independent with uncorrelated uncertainties."

**The following sentence is also added in section 3 line 153 to clarify how the trend confidence limits are calculated:** "The trend uncertainties are calculated using the 95% confidence limit of the slope of the linear regression assuming that the residuals are not correlated for weekly (ECC) or 2/3 per week (lidar) observations."

*The persistent 5% bias compared to total ozone measurements is larger than is seen at most stations, and quite surprising (and disappointing) after thorough homogenization. I think it deserves more discussion, perhaps in the context of Dr. Stauffer's recent work, or JOSIE results. Are other stations just lucky, or are there undiagnosed problems with the OHP time series?*

**The total ozone measurement calculation method are now identical in section 4.1 and 4.3 and the negative bias with OHP SAOZ total ozone measurement decreases down to -3.7% (Fig. 4). This bias is also smaller (-1%±2%) for the comparison with satellite TOC observations (Fig. 8 now 9). Comparison with LiO3St is now also made with a larger data set of 366 sondes launched within 12 hours of the LiO3St observations during the 2005-2021 observing period of MLS (see new figure 6). For this data set ECC-LiO3St is only -2% in the 22-26 km altitude, while ECC-MLS and ECC-LiO3St do not have the same sign after the homogenization. So the homogenization did improve the data quality in the stratosphere. The following paragraphs are now included in section 4.2 line 217:** " For the period 2005-2021 and using a time difference less than 12 hours, the negative bias between the homogenized ECC and the lidar decreases down to -2% between 22 and 24 km, but remains as large as -7% above 28 km (Fig.6b)."

**and section 4.3 line 258** "The fact that the average ECC-MLS difference shown in Fig. 6b is slightly positive (+2%) in the 22-26 km altitude range, while the average ECC-LiO3St difference is slightly negative (-2%) means that homogenization is a good compromise for intercomparability with other techniques measuring $O_3$ in the stratosphere below 26 km. Above 26 km, both comparisons indicate a negative bias in homogenized ECC $O_3$ concentrations of less than -6%."

**The following paragraph has been added in the conclusion line 359 to compare our results with the benefits of homogenization performed in the SHADOZ network and in the Uccle/De Bilt sites in Europe:** "While the objective of this paper is to discuss the impact of homogenization on the OHP dataset using lidar and satellite measurements, it is worth checking how such corrections have improved data quality at other sites. The impact of the homogenization is dependent on the site, because different homogenization steps have to be applied at different stations. In general, the additional corrections for the pump temperature will give higher ozone partial pressure amounts in the stratosphere. On the other hand, applying a constant background current subtraction instead of a pressure dependent background current and applying the transfer functions from EnSci-SST 1% will lead to lower ozone partial pressure values above 10 km. Witte et al. (2017) performed an extensive analysis of 7 SHADOZ network stations in the tropics, showing that the mean differences between ECC and MLS are reduced from -11.2±13.6% to -3.0±10% at 40 hPa (22 km) and from -3.2±4% to -0.7±3.1% at 17 hPa (28 km). In Europe, Van Malderen et al. (2016) observed that the O3S-DQA corrections actually give higher (+1%) and lower (-2%) ozone concentrations in the stratosphere with respect to standard processing for the Uccle 1997-2014 and De Bilt 1993-2014 ECC observations, respectively. This is mainly due to the fact that the pump temperature correction was a major correction for Uccle, while changing the background current correction has a major effect for De Bilt. O3S-DQA corrections reduce the relative $O_3$ difference

between Uccle and De Bilt in the lower stratosphere. The analysis of homogenized ECC at OHP using LiO3St or MLS show similar improvements in the stratosphere below 26 km. The remaining bias of -2% to -3.7% between homogenized ECC and other techniques measuring $O_3$ in the stratosphere at OHP is also in the range of the remaining negative differences between homogenized ECC and MLS observed in the 22 to 28 km altitude range by 4 stations of the SHADOZ network (Witte et al., 2017).}"

*Minor points:*

*Section 2: Were the OHP ECC data ever normalized to a total ozone measurement? (The Brewer-Mast data would have been.) In any case it would be worth stating here explicitly that the homogenized data are NOT normalized, even though a normalization factor is calculated.*

**This is correct there is no normalization. This is now said explicitly in section 2 line 97.** "The homogenized data are not normalized with this normalization factor which is only used as a quality flag".

*Line 60: "No more vertical smoothing of the ozone partial pressure". This seems to imply that this was done before. Correct?*

**Yes it was. Text is now** "No more vertical smoothing of the ozone partial pressure \red{while smoothing over 100 m was appied in the uncorrected data"

*Line 67: "Only Komhyr86 is applied for the current to PO3 conversion of uncorrected data." Do you mean that Komhyr86 was used for both sonde types previously?*

**Yes. Text is now** "Komhyr86 was applied for the current to $P_{O3}$ conversion of all the uncorrected data."

*Line 69-70: No, we don't see that. There are too many points, many of them overlapping, to tell by visual inspection whether there is a trend. Please put a regression line through them.*

**The regression line would not be very useful here. Text has been changed in section 2 line 71**: "The comparison of $I_b$ used before and after homogenization is shown in Fig.1. The standard deviation of the background current between 1991 and 2021 remains on the order of ±0.05 and only 17% of the $I_b$ values are greater than the mean of the uncorrected $I_b$ after homogenization."

*Lines 96-99: How is the uncertainty calculated? At least a brief description is necessary, and/or a reference to a comprehensive description.*

**Reference to Smit et al. (2021) has been added. Text in section 2 line 103 is now:** " The detailed description of the uncertainty calculation is given in \cite{Smit2021}. All the error terms have been included in our calculation except the bias due the sensor time response and the pressure uncertainty."

*Line 117: I presume this is the correction proposed by the BIPM (which is now 1.23%, not the 1.8% suggested originally).*

**It is now said less than 2%**

*Line 155: "normalization factor NT": this should be defined.*

**It is defined in section 2 line 93**

*Line 174: "...assuming an independent error for the 40 comparisons taken into account." Do you mean "...assuming that the 40 comparisons were independent, with uncorrelated errors." ?*

**Yes Text was changed (see answer to major comment #1)**

*Line 178: "It is due to the difference..." These are processing differences --- not really errors, especially since the assumption of a constant background current is also wrong (e.g. ASOPOS 2.0 report).*

**Yes we agree, wording is changed section 4.2 line 96:** "It may be explained by differences introduced by not correcting the $O_3$ partial pressure for EnSci-SST 1% and by using a pressure dependent background current subtraction".

*Line 228-229: "No post 2013 drop off in TOC measurement by the ECC is seen at OHP as observed at other measurement sites in Stauffer et al. (2020). " Really? From your own Figure 8, I'd estimate the dropoff at about 2%.*

**Text has been changed in section 4.3 line 267 by:** "A small post 2013 drop-off in TOC measurement of -2% by the ECC at OHP might be present, but is considerably less prominent than the drop-off observed at other measurement sites in Stauffer et al. (2020)"

*Figure 8: Is the difference sonde-satellite or satellite-sonde?*

**Text line 224 (now 263) was correct but caption of Fig.8 (now 9) was indeed wrong. Fig.9 caption is corrected.**

*Line 230-231: "The differences are mostly negative and between -4% and 1% after homogenization. So the satellite/ECC TOC comparison is consistent with the time distribution of the normalization factor shown in Fig.4." Not*

*really --- the normalization factor averages 5% difference. I agree that Figure 8 (lower) looks pretty good. So what's with Figure 4?*

**The difference in TOC between the ECC and the satellite is on the order of -1%±2%, thus effectively less than the average difference between the ECC and OHP TOC observations. As explained in response to major comment #2, the difference is less with the new ECC TOC calculation in section 4.1. The following sentences are now included in Section 4.3 line 270:** " The ECC minus satellite TOC temporal evolution is consistent with the time distribution of the normalization factor shown in Fig.4. However TOC differences are close to zero between 2010 and 2016 using the satellite data, while a -3% bias is present using the OHP total ozone measurements. In this context, we mention that the expected bias between GOME and SAOZ is between -3% to +1% (Hendrick et al., 2011)"

**and in the conclusion line 347:** "Differences between TOC measured by ECC and by GOME or OMI/OMPS switch from 2%±2% for uncorrected ECC to -1%±2% for homogenized ECC. The negative bias is then smaller than the -3.7\% obtained with the OHP TOC measurements, eventhough the time evolution is consistent with the $N_T$ time distribution"